# Ordered Memory

**Yikang Shen**[*]
Mila/Université de Montréal
and Microsoft Research
Montréal, Canada

**Shawn Tan**[*]
Mila/Université de Montréal
Montréal, Canada

**Arian Hosseini**[*]
Mila/Université de Montréal
and Microsoft Research
Montréal, Canada

**Zhouhan Lin**
Mila/Université de Montréal
Montréal, Canada

**Alessandro Sordoni**
Microsoft Research
Montréal, Canada

**Aaron Courville**
Mila/Université de Montréal
Montréal, Canada

## Abstract

Stack-augmented recurrent neural networks (RNNs) have been of interest to the deep learning community for some time. However, the difficulty of training memory models remains a problem obstructing the widespread use of such models. In this paper, we propose the Ordered Memory architecture. Inspired by Ordered Neurons (Shen et al., 2018), we introduce a new attention-based mechanism and use its cumulative probability to control the writing and erasing operation of memory. We also introduce a new Gated Recursive Cell to compose lower level representations into higher level representation. We demonstrate that our model achieves strong performance on the logical inference task (Bowman et al., 2015) and the ListOps (Nangia and Bowman, 2018) task. We can also interpret the model to retrieve the induced tree structure, and find that these induced structures align with the ground truth. Finally, we evaluate our model on the Stanford Sentiment Treebank tasks (Socher et al., 2013), and find that it performs comparatively with the state-of-the-art methods in the literature[2].

## 1 Introduction

A long-sought after goal in natural language processing is to build models that account for the compositional nature of language — granting them an ability to understand complex, unseen expressions from the meaning of simpler, known expressions (Montague, 1970; Dowty, 2007). Despite being successful in language generation tasks, recurrent neural networks (RNNs, Elman (1990)) fail at tasks that explicitly require and test compositional behavior (Lake and Baroni, 2017; Loula et al., 2018). In particular, Bowman et al. (2015), and later Bahdanau et al. (2018) give evidence that, by exploiting the appropriate compositional structure of the task, models can generalize better to out-of-distribution test examples. Results from Andreas et al. (2016) also indicate that recursively composing smaller modules results in better representations. The remaining challenge, however, is learning the underlying structure and the rules governing composition from the observed data alone. This is often referred to as the *grammar induction* (Chen, 1995; Cohen et al., 2011; Roark, 2001; Chelba and Jelinek, 2000; Williams et al., 2018).

Fodor and Pylyshyn (1988) claim that "cognitive capacities always exhibit certain symmetries, so that the ability to entertain a given thought implies the ability to entertain thoughts with semantically related contents," and use the term *systematicity* to describe this phenomenon. Exploiting

---

[*]yi-kang.shen@umontreal.ca, tanjings@mila.quebec, arian.hosseini9@gmail.com.
[2]The code can be found at https://github.com/yikangshen/Ordered-Memory

known symmetries in the structure of the data has been a useful technique for achieving good generalization capabilities in deep learning, particularly in the form of convolutions (Fukushima, 1980), which leverage parameter-sharing. If we consider architectures used in Socher et al. (2013) or Tai et al. (2015), the same recursive operation is performed at known points along the input where the substructures are meant to be composed. Could symmetries in the structure of natural language data be learned and exploited by models that operate on them?

In recent years, many attempts have been made in this direction using neural network architectures (Grefenstette et al., 2015; Bowman et al., 2016; Williams et al., 2018; Yogatama et al., 2018; Shen et al., 2018; Dyer et al., 2016). These models typically augment a recurrent neural network with a stack and a buffer which operate in a similar way to how a shift-reduce parser builds a parse-tree. While some assume that ground-truth trees are available for supervised learning (Bowman et al., 2016; Dyer et al., 2016), others use reinforcement learning (RL) techniques to learn the optimal sequence of shift reduce actions in an unsupervised fashion (Yogatama et al., 2018).

To avoid some of the challenges of RL training (Havrylov et al., 2019), some approaches use a *continuous stack* (Grefenstette et al., 2015; Joulin and Mikolov, 2015; Yogatama et al., 2018). These can usually only perform one push or pop action per time step, requiring different mechanisms — akin to adaptive computation time (ACT, Graves (2016); Jernite et al. (2016)) — to perform the right number of shift and reduce steps to express the correct parse. In addition, continuous stack models tend to "blur" the stack due to performing a "soft" shift of either the pointer to the head of the stack (Grefenstette et al., 2015), or all the values in the stack (Joulin and Mikolov, 2015; Yogatama et al., 2018). Finally, while these previous models can learn to manipulate a stack, they lack the capability to *lookahead* to future tokens before performing the stack manipulation for the current time step.

In this paper, we propose a novel architecture: *Ordered Memory* (OM), which includes a new memory updating mechanism and a new Gated Recursive Cell. We demonstrate that our method generalizes for synthetic tasks where the ability to parse is crucial to solving them. In the Logical inference dataset (Bowman et al., 2015), we show that our model can systematically generalize to unseen combination of operators. In the ListOps dataset (Nangia and Bowman, 2018), we show that our model can learn to solve the task with an order of magnitude less training examples than the baselines. The parsing experiments shows that our method can effectively recover the latent tree structure of the both tasks with very high accuracy. We also perform experiments on the Stanford Sentiment Treebank, in both binary classification and fine-grained settings (SST-2 & SST-5), and find that we achieve comparative results to the current benchmarks.

## 2 Related Work

Composition with recursive structures has been shown to work well for certain types of tasks. Pollack (1990) first suggests their use with distributed representations. Later, Socher et al. (2013) shows their effectiveness on sentiment analysis tasks. Recent work has demonstrated that recursive composition of sentences is crucial to systematic generalisation (Bowman et al., 2015; Bahdanau et al., 2018). Kuncoro et al. (2018) also demonstrate that architectures like Dyer et al. (2016) handle syntax-sensitive dependencies better for language-related tasks.

Schützenberger (1963) first showed an equivalence between push-down automata (stack-augmented automatons) and context-free grammars. Knuth (1965) introduced the notion of a shift-reduce parser that uses a stack for a subset of formal languages that can be parsed from left to right. This technique for parsing has been applied to natural language as well: Shieber (1983) applies it to English, using assumptions about how native English speakers parse sentences to remove ambiguous parse candidates. More recently, Maillard et al. (2017) shows that a soft tree could emerge from all possible tree structures through back propagation.

The idea of using neural networks to control a stack is not new. Zeng et al. (1994) uses gradient estimates to learn to manipulate a stack using a neural network. Das et al. (1992) and Mozer and Das (1993) introduced the notion of a *continuous stack* in order for the model to be fully differentiable. Much of the recent work with stack-augmented networks built upon the development of neural attention (Graves, 2013; Bahdanau et al., 2014; Weston et al., 2014). Graves et al. (2014) proposed methods for reading and writing using a head, along with a "soft" shift mechanism. Apart from using attention mechanisms, Grefenstette et al. (2015) proposed a neural stack where the push

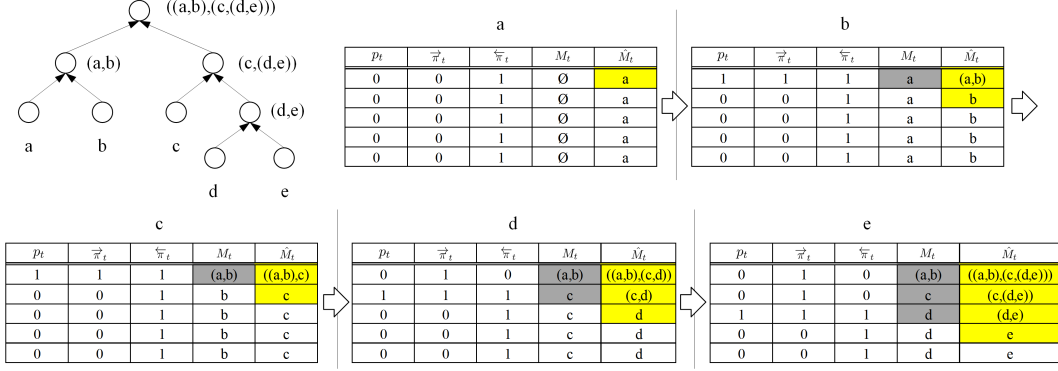

Tables for a and b:

| $p_t$ | $\vec{\pi}_t$ | $\overleftarrow{\pi}_t$ | $M_t$ | $\hat{M}_t$ |
|---|---|---|---|---|
| 0 | 0 | 1 | Ø | a |
| 0 | 0 | 1 | Ø | a |
| 0 | 0 | 1 | Ø | a |
| 0 | 0 | 1 | Ø | a |
| 0 | 0 | 1 | Ø | a |

| $p_t$ | $\vec{\pi}_t$ | $\overleftarrow{\pi}_t$ | $M_t$ | $\hat{M}_t$ |
|---|---|---|---|---|
| 1 | 1 | 1 | a | (a,b) |
| 0 | 0 | 1 | a | b |
| 0 | 0 | 1 | a | b |
| 0 | 0 | 1 | a | b |
| 0 | 0 | 1 | a | b |

Tables for c, d, e:

| $p_t$ | $\vec{\pi}_t$ | $\overleftarrow{\pi}_t$ | $M_t$ | $\hat{M}_t$ |
|---|---|---|---|---|
| 1 | 1 | 1 | (a,b) | ((a,b),c) |
| 0 | 0 | 1 | b | c |
| 0 | 0 | 1 | b | c |
| 0 | 0 | 1 | b | c |
| 0 | 0 | 1 | b | c |

| $p_t$ | $\vec{\pi}_t$ | $\overleftarrow{\pi}_t$ | $M_t$ | $\hat{M}_t$ |
|---|---|---|---|---|
| 0 | 1 | 0 | (a,b) | ((a,b),(c,d)) |
| 1 | 1 | 1 | c | (c,d) |
| 0 | 0 | 1 | c | d |
| 0 | 0 | 1 | c | d |
| 0 | 0 | 1 | c | d |

| $p_t$ | $\vec{\pi}_t$ | $\overleftarrow{\pi}_t$ | $M_t$ | $\hat{M}_t$ |
|---|---|---|---|---|
| 0 | 1 | 0 | (a,b) | ((a,b),(c,(d,e))) |
| 0 | 1 | 0 | c | (c,(d,e)) |
| 1 | 1 | 1 | d | (d,e) |
| 0 | 0 | 1 | d | e |
| 0 | 0 | 1 | d | e |

Figure 1: An example run of the OM model. Let the input sequence $a, b, c, d, e$ and its hierarchical structure be as shown in the figure. Ideally, the OM model will output the values shown in the above tables. The occupied slots in $M_t$ are highlighted in gray. The yellow slots in $\hat{M}_t$ are slots that can be attended on in time-step $t + 1$. At the first time-step ($t = 1$), the model will initialize the candidate memory $\hat{M}_1$ with input $a$ and the memory $M_0$ with zero vectors. At $t = 2$, the model attends on the last memory slot to compute $M_1$ (Eqn. 5), followed by $\hat{M}_2$ (Eqn. 7). At $t = 3$, given the input $c$, the model will attend on the last slot. Consequently the memory slot for $b$ is erased by $\vec{\pi}_3$. Given Eqns. 6 and 7, our model will recursively compute every slot in the candidate memory $\hat{M}_t^i$ to include information from $\hat{M}_t^{i-1}$ and $M_{t-1}^i$. Since the cell($\cdot$) function only takes 2 inputs, the actual computation graph is a binary tree.

and pop operations are made to be differentiable, which worked well in synthetic datasets. Yogatama et al. (2016) proposes RL-SPINN where the discrete stack operations are directly learned by reinforcement learning.

## 3 Model

The OM model actively maintains a stack and processes the input from left to right, with a one-step lookahead in the sequence. This allows the OM model to decide the local structure more accurately, much like a shift-reduce parser (Knuth, 1965).

At a given point $t$ in the input sequence $\boldsymbol{x}$ (the $t$-th time-step), we have a memory of candidate sub-trees spanning the non-overlapping sub-sequences in $x_1, \ldots, x_{t-1}$, with each sub-tree being represented by one slot in the memory stack. We also maintain a memory stack of sub-trees that contains $x_1, \ldots, x_{t-2}$. We use the input $x_t$ to choose its parent node from our previous candidate sub-trees. The descendant sub-trees of this new sub-tree (if they exist) are removed from the memory stack, and this new sub-tree is then added. We then build the new candidate sub-trees that include $x_t$ using the current input and the memory stack. In what follows, we describe the OM model in detail. To facilitate a clearer description, a discrete attention scheme is assumed, but only "soft" attention is used in both the training and evaluation of this model.

Let $D$ be the dimension of each memory slot and $N$ be the number of memory slots. At time-step $t$, the model takes four inputs:

- $M_{t-1}$: a memory matrix of dimension $N \times D$, where each occupied slot is a distributed representation for sub-trees spanning the non-overlapping subsequences in $x_1, ..., x_{t-2}$;

- $\hat{M}_{t-1}$: a matrix of dimension $N \times D$ that contains representations for candidate subtrees that include the leaf node $x_{t-1}$;

- $\vec{\pi}_{t-1}$: a vector of dimension $N$, where each element indicate whether the respective slot in $M_{t-1}$ occupied by a subtree.

- $x_t$: a vector of dimension $D_{in}$, the input at time-step $t$.

The model first transforms $x_t$ to a $D$ dimension vector.

$$\tilde{x}_t = LN(W x_t + b) \tag{1}$$

where $LN(\cdot)$ is the layer normalization function (Ba et al., 2016).

To select the candidate representations from $\hat{M}_{t-1}$, the model uses $\tilde{x}_t$ as its query to attend on $\hat{M}_{t-1}$:

$$p_t = \text{Att}(\tilde{x}_t, \hat{M}_{t-1}, \overrightarrow{\pi}_{t-1}) \qquad (2)$$

$$\overrightarrow{\pi}_t^i = \sum_{j \leq i} p_t^j \qquad (3)$$

$$\overleftarrow{\pi}_t^i = \sum_{j \geq i} p_t^j \qquad (4)$$

where $\text{Att}(\cdot)$ is a masked attention function, $\overrightarrow{\pi}_{t-1}$ is the mask, $p_t$ is a distribution over different memory slots in $\hat{M}_{t-1}$, and $p_t^j$ is the probability on the $j$-th slot. The attention mechanism will be described in section 3.1. Intuitively, $p_t$ can be viewed as a pointer to the head of the stack, $\overrightarrow{\pi}_t$ is an indicator value over where the stack exists, and $\overleftarrow{\pi}_t$ is an indicator over where the top of the stack is and where it is non-existent.

To compute $M_t$, we combine $\hat{M}_{t-1}$ and $M_{t-1}$ through:

$$M_t^i = M_{t-1}^i \cdot (1 - \overleftarrow{\pi})^i + \hat{M}_{t-1}^i \cdot \overleftarrow{\pi}_t^i, \quad \forall i \qquad (5)$$

Suppose $p_t$ points at a memory slot $y_t$ in $\hat{m}$. Then $\overleftarrow{\pi}_t$ will write $\hat{M}_{t-1}^i$ to $M_t^i$ for $i \leq y_t$, and $(1 - \overleftarrow{\pi}_t)$ will write $M_{t-1}^i$ to $M_t^i$ for $i > y_t$. In other words, Eqn. 5 copies everything from $M_{t-1}$ to the current timestep, up to the where $p_t$ is pointing.

We believe that this is a crucial point that differentiates our model from past stack-augmented models like Yogatama et al. (2016) and Joulin and Mikolov (2015). Both constructions have the 0-th slot as the top of the stack, and perform a convex combination of each slot in the memory / stack given the action performed. More concretely, a distribution over the actions that is not sharp (e.g. 0.5 for pop) will result in a weighted sum of an un-popped stack and a pop stack, resulting in a blurred memory state. Compounded, this effect can make such models hard to train. In our case, because $(1 - \overleftarrow{\pi}_t)^i$ is non-decreasing with $i$, its value will accumulate to 1 at or before $N$. This results in a full copy, guaranteeing that the earlier states are retained. This full retention of earlier states may play a part in the training process, as it is a strategy also used in Gulcehre et al. (2017), where all the memory slots are filled before any erasing or writing takes place.

**Data:** $x_1, ..., x_T$
**Result:** $o_T^N$
initialize $M_0, \hat{M}_0$;
**for** $i \leftarrow 1$ **to** $T$ **do**
    $\tilde{x}_t = LN(Wx_t + b)$;
    $p_t = \text{Att}(\tilde{x}_t, \hat{M}_{t-1}, \overrightarrow{\pi}_{t-1})$;
    $\overrightarrow{\pi}_t^i = \sum_{j \leq i} p_t^j$;
    $\overleftarrow{\pi}_t^i = \sum_{j \geq i} p_t^j$;
    $\hat{M}_t^0 = \tilde{x}_t$;
    **for** $i \leftarrow 1$ **to** $N$ **do**
        $M_t^i = M_{t-1}^i \cdot (1 - \overleftarrow{\pi}_t)^i + \hat{M}_{t-1}^i \cdot \overleftarrow{\pi}_t^i$;
        $o_t^i = \text{cell}(M_t^i, \hat{M}_t^{i-1})$;
        $\hat{M}_t^i = \tilde{x}_t \cdot (1 - \overrightarrow{\pi}_t)^i + o_t^i \cdot \overrightarrow{\pi}_t^i$;
    **end**
**end**
**return** $o_T^N$;

**Algorithm 1:** Ordered Memory algorithm. The attention function $\text{Att}(\cdot)$ is defined in section 3.1. The recursive cell function $\text{cell}(\cdot)$ is defined in section 3.2.

To compute candidate memories for time step $t$, we recurrently update all memory slots with

$$o^i = \text{cell}(M_t^i, \hat{M}_t^{i-1}) \qquad (6)$$

$$\hat{M}_t^i = \tilde{x}_t \cdot (1 - \overrightarrow{\pi}_t)^{i+1} + o_t^i \cdot \overrightarrow{\pi}_t^i, \forall i \qquad (7)$$

where $\hat{M}_t^0$ is $x_t$. The $\text{cell}(\cdot)$ function can be seen as a recursive composition function in a recursive neural network (Socher et al., 2013). We propose a new cell function in section 3.2.

The output of time step $t$ is the last memory slot $\hat{M}_t^N$ of the new candidate memory, which summarizes all the information from $x_1, ..., x_t$ using the induced structure. The pseudo-code for the OM algorithm is shown in Algorithm 1.

### 3.1 Masked Attention

Given the projected input $\tilde{x}_t$ and candidate memory $\hat{M}_{t-1}^i$. We feed every $(\tilde{x}_t, \hat{M}_{t-1}^i)$ pair into a feed-forward network:

$$\alpha_t^i = \frac{\mathbf{w}_2^{Att} \tanh\left(\mathbf{W}_1^{Att} \begin{bmatrix} \hat{M}_{t-1}^i \\ \tilde{x}_t \end{bmatrix} + b_1 \right) + b_2}{\sqrt{N}} \tag{8}$$

$$\beta_t^i = \exp\left(\alpha_t^i - \max_j \alpha_t^j \right) \tag{9}$$

where $\mathbf{W}_1^{Att}$ is $N \times 2N$ matrix, $\mathbf{w}_2^{Att}$ is $N$ dimension vector, and the output $\beta_t^i$ is a scalar. The purpose of dividing by $\sqrt{N}$ is to scale down the logits before softmax is applied, a technique similar to the one seen in Vaswani et al. (2017). We further mask the $\beta_t$ with the cumulative probability from the previous time step to prevent the model attending on non-existent parts of the stack:

$$\hat{\beta}_t^i = \beta_t^i \overrightarrow{\pi}_{t-1}^{i+1} \tag{10}$$

where $\overrightarrow{\pi}_{t-1}^{N+1} = 1$ and $\overrightarrow{\pi}_0^{\leq N} = 0$. We can then compute the probability distribution:

$$p_t^i = \frac{\hat{\beta}_t^i}{\sum_j \hat{\beta}_t^j} \tag{11}$$

This formulation bears similarity to the method used for the multi-pop mechanism seen in Yogatama et al. (2018).

### 3.2 Gated Recursive Cell

Instead of using the recursive cell proposed in TreeLSTM (Tai et al., 2015) and RNTN (Socher et al., 2010), we propose a new gated recursive cell, which is inspired by the feed-forward layer in Transformer (Vaswani et al., 2017). The inputs $M_t^i$ and $\hat{M}_t^{i-1}$ are concatenated and fed into a fully connect feed-forward network:

$$\begin{bmatrix} v_t^i \\ h_t^i \\ c_t^i \\ u_t^i \end{bmatrix} = \mathbf{W}_2^{Cell} \text{ReLU}\left(\mathbf{W}_1^{Cell} \begin{bmatrix} \hat{M}_t^{i-1} \\ M_t^i \end{bmatrix} + b_1 \right) + b_2 \tag{12}$$

Like the TreeLSTM, we compute the output with a gated combination of the inputs and $u_t^i$:

$$o_t^i = LN(\sigma(v_t^i) \odot \hat{M}_t^{i-1} + \sigma(h_t^i) \odot M_t^i + \sigma(c_t^i) \odot u_t^i) \tag{13}$$

where $v_t^i$ is the vertical gate that controls the input from previous slot, $h_t^i$ is horizontal gate that controls the input from previous time step, $cg_t^i$ is cell gate that control the $u_t^i$, $o_t^i$ is the output of cell function, and $LN(\cdot)$ share the same parameters with the one used in the Eqn. 1.

### 3.3 Relations to ON-LSTM and Shift-reduce Parser

Ordered Memory is implemented following the principles introduced in Ordered Neurons (Shen et al., 2018). Our model is related to ON-LSTM in several aspects: 1) The memory slots are similar to the chunks in ON-LSTM, when a higher ranking memory slot is forgotten/updated, all lower ranking memory slots should likewise be forgotten/updated; 2) ON-LSTM uses the monotonically non-decreasing master forget gate to preserve long-term information while erasing short term information, the OM model uses the cumulative probability $\overrightarrow{\pi}_t$; 3) Similarly, the master input gate used by ON-LSTM to control the writing of new information into the memory is replaced with the reversed cumulative probability $\overleftarrow{\pi}_t$ in the OM model.

At the same time, the internal mechanism of OM can be seen as a continuous version of a shift-reduce parser. At time step $t$, a shift-reduce parser could perform zero or several reduce steps to combine the heads of stack, then shift the word $t$ into stack. The OM implement the reduce step with Gated Recursive Cell. It combines $\hat{M}_t^{i-1}$, the output of previous reduce step, and $M_t^i$,

Table 1: Test accuracy of the models, trained on operation lengths of $\leq 6$, with their out-of-distribution results shown here (lengths 7-12). We ran 5 different runs of our models, giving the error bounds in the last row. The $F_1$ score is the parsing score with respect to the ground truth tree structure. The TreeCell is a recursive neural network based on the Gated Recursive Cell function proposed in section 3.2. For the Transformer and Universal Transformer, we follow the entailment architecture introduced in Radford et al. (2018). The model takes `<start> sentence1 <delim> sentence2 <extract>` as input, then use the vector representation for `<extract>` position at last layer for classification. *The results for RRNet were taken from Jacob et al. (2018).

| Model | Number of Operations | | | | | | Sys. Gen. | | |
|---|---|---|---|---|---|---|---|---|---|
| | 7 | 8 | 9 | 10 | 11 | 12 | A | B | C |
| *Sequential sentence representation* | | | | | | | | | |
| LSTM | 88 | 84 | 80 | 78 | 71 | 69 | 84 | 60 | 59 |
| RRNet* | 84 | 81 | 78 | 74 | 72 | 71 | – | – | – |
| ON-LSTM | 91 | 87 | 85 | 81 | 78 | 75 | 70 | 63 | 60 |
| *Inter sentence attention* | | | | | | | | | |
| Transformer | 51 | 52 | 51 | 51 | 51 | 48 | 53 | 51 | 51 |
| Universal Transformer | 51 | 52 | 51 | 51 | 51 | 48 | 53 | 51 | 51 |
| *Our model* | | | | | | | | | |
| Accuracy | $98 \pm 0.0$ | $97 \pm 0.4$ | $96 \pm 0.5$ | $94 \pm 0.8$ | $93 \pm 0.5$ | $92 \pm 1.1$ | 94 | 91 | 81 |
| Parsing $F_1$ | | | $84.3 \pm 14.4$ | | | | | | |
| *Ablation tests* | | | | | | | | | |
| ~~cell(-)~~ TreeRNN Op. | 69 | 67 | 65 | 61 | 57 | 53 | – | – | – |
| *Recursive NN + ground-truth structure* | | | | | | | | | |
| TreeLSTM | 94 | 92 | 92 | 88 | 87 | 86 | 91 | 84 | 76 |
| TreeCell | 98 | 96 | 96 | 95 | 93 | 92 | 95 | 95 | 90 |
| TreeRNN | 98 | 98 | 97 | 96 | 95 | 96 | 94 | 92 | 86 |

the next element in stack, into $\hat{M}_t^i$, the representation for new sub-tree. The number of reduce steps is modeled with the attention mechanism. The probability distribution $p_t$ models the position of the head of stack after all necessary reduce operations are performed. The shift operations is implemented as copying the current input word $x_t$ into memory.

The upshot of drawing connections between our model and the shift-reduce parser is interpretability: We can approximately infer the computation graph constructed by our model with Algorithm 2 (see appendix). The algorithm can be used for the latent tree induction tasks in (Williams et al., 2018).

## 4 Experiments

We evaluate the tree learning capabilities of our model on two datasets: logical inference (Bowman et al., 2015) and ListOps (Nangia and Bowman, 2018). In these experiments, we infer the trees with our model by using Alg. 2 and compare them with the ground-truth trees used to generate the data. We evaluate parsing performance using the $F_1$ score[3]. We also evaluate our model on Stanford Sentiment Treebank (SST), which is the sequential labeling task described in Socher et al. (2013).

### 4.1 Logical Inference

The logical inference task described in Bowman et al. (2015) has a vocabulary of six words and three logical operations, `or`, `and`, `not`. The task is to classify the relationship of two logical clauses into seven mutually exclusive categories. We use a multi-layer perceptron (MLP) with $(h_1, h_2, h_1 \circ h_2, |h_1 - h_2|)$ as input, where $h_1$ and $h_2$ are the $\hat{M}_T^N$ of their respective input sequences. We compare our model with LSTM, RRNet (Jacob et al., 2018), ON-LSTM (Shen et al., 2018), Tranformer (Vaswani et al., 2017), Universal Transformer (Dehghani et al., 2018), TreeLSTM (Tai et al., 2015), TreeRNN (Bowman et al., 2015), and TreeCell. We used the same hidden state size for our model and baselines for proper comparison. Hyper-parameters can be found in Appendix B. The model

Table 2: Partitions of the Logical Inference task from Bowman et al. (2014). Each partitions include a training set filtered out all data points that match the rule indicated in **Excluded**, and a test set formed by matched data points.

| Part. | Excluded | Training set size | Test set example |
|---|---|---|---|
| A | `* ( and (not a) ) *` | 128,969 | `f (and (not a))` |
| B | `* ( and (not *) ) *` | 87,948 | `c (and (not (a (or b))))` |
| C | `* ( {and,or} (not *) ) *` | 51,896 | `a (or (e (and c)))` |
| Full | | 135,529 | |

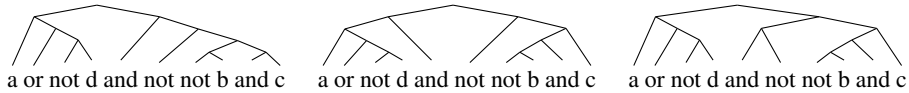

a or not d and not not b and c    a or not d and not not b and c    a or not d and not not b and c

Figure 2: Variations in induced parse trees under different runs of the logical inference experiment. The left most tree is the ground truth and one of induced structures. We have removed the parentheses in the original sequence for this visualization. It is interesting to note that the different structures induced by our model are all valid computation graphs to produce the correct results.

is trained on sequences containing up to 6 operations and tested on sequences with higher number (7-12) of operations.

The Transformer models were implemented by modifying the code from the Annotated Transformer[4]. The number of Transformer layers are the same as the number of slots in Ordered Memory. Unfortunately, we were not able to successfully train a Transformer model on the task, resulting in a model that only learns the marginal over the labels. We also tried to used Transformer as a sentence embedding model, but to no avail. Tran et al. (2018) achieves similar results, suggesting this could be a problem intrinsic to self-attention mechanisms for this task.

**Length Generalization Tests** The TreeRNN model represents the best results achievable if the structure of the tree is known. The TreeCell experiment was performed as a control to isolate the performance of using the $\text{cell}(\cdot)$ composition function versus using both using $\text{cell}(\cdot)$ and learning the composition with OM. The performance of our model degrades only marginally with increasing number of operations in the test set, suggesting generalization on these longer sequences never seen during training.

**Parsing results** There is a variability in parsing performance over several runs under different random seeds, but the model's ability to generalize to longer sequences remains fairly constant. The model learns a slightly different method of composition for consecutive operations. Perhaps predictably, these are variations that do not affect the logical composition of the subtrees. The source of different parsing results can be seen in Figure 2. The results suggest that these latent structures are still valid computation graphs for the task, in spite of the variations.

**Systematic Generalization Tests** Inspired by Loula et al. (2018), we created three splits of the original logical inference dataset with increasing levels of difficulty. Each consecutive split removes a superset of the previously excluded clauses, creating a harder generalization task. Each model is then trained on the ablated training set, and tested on examples unseen in the training data. As a result, the different test splits have different numbers of data points. Table 2 contains the details of the individual partitions.

The results are shown in the right section of Table 1 under Sys. Gen. Each column labeled A, B, and C are the model's aggregated accuracies over the unseen operation lengths. As with the length generalization tests, the models with the known tree structure performs the best on unseen structures, while sequential models degrade quickly as the tests get harder. Our model greatly outperforms all the other sequential models, performing slightly below the results of TreeRNN and TreeCell on the different partitions.

| Model | Accuracy | $F_1$ |
|---|---|---|
| *Baselines* | | |
| LSTM* | 71.5±1.5 | – |
| RL-SPINN* | 60.7±2.6 | 71.1 |
| Gumbel Tree-LSTM* | 57.6±2.9 | 57.3 |
| Transformer | 57.4±0.4 | – |
| Universal Transformer | 71.5±7.8 | – |
| Havrylov et al. (2019) | 99.2±0.5 | – |
| Ours | **99.97±0.014** | **100** |
| *Ablation tests* | | |
| ~~cell(·)~~ TreeRNN Op. | 63.1 | – |

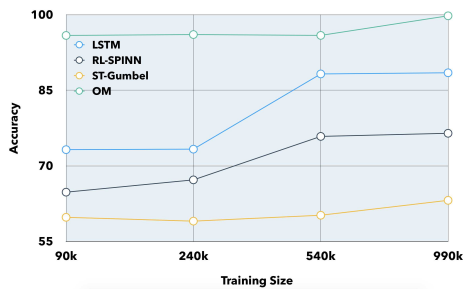

(a)                    (b)

Figure 3: (a) shows the accuracy of different models on the ListOps dataset. All models have 128 dimensions. Results for models with * are taken from Nangia and Bowman (2018). (b) shows our model accuracy on the ListOps task when varying the the size of the training set.

Combined with the parsing results, and our model's performance on these generalization tests, we believe this is strong evidence that the model has both (i) learned to exploit symmetries in the structure of the data by learning a good cell(·) function, and (ii) learned where and how to apply said function by operating its stack memory.

## 4.2 ListOps

Nangia and Bowman (2018) build a dataset with nested summary operations on lists of single digit integers. The sequences comprise of the operators MAX, MIN, MED, and SUM_MOD. The output is also an integer in $[0, 9]$ As an example, the input: [MAX 2 9 [MIN 4 7 ] 0 ] has the solution 9. As the task is formulated to be easily solved with a correct parsing strategy, the task provides an excellent test-bed to diagnose models that perform tree induction. The authors binarize the structure by choosing the subtree corresponding to each list to be left-branching: the model would first take into account the operator, and then proceed to compute the summary statistic within the list. A right-branching parse would require the entire list to be maintained in the model's hidden state.

Our model achieves 99.9% accuracy, and an $F_1$ score of 100% on the model's induced parse tree (See Table 3a). This result is consistent across 3 different runs of the same experiment. In Nangia and Bowman (2018), the authors perform an experiment to verify the effect of training set size on the latent tree models. As the latent tree models (RL-SPINN and ST-Gumbel) need to parse the input successfully to perform well on the task, the better performance of the LSTM than those models indicate that the size of the dataset does not affect the ability to learn to parse much for those models. Our model seems to be more data efficient and solves the task even when only training on a subset of 90k examples (Fig. 3b).

## 4.3 Ablation studies

We replaced the cell(·) operator with the RNN operator found in TreeRNN, which is the best performing model that explicitly uses the structure of the logical clause. In this test, we find that the TreeRNN operator results in a large drop across the different tasks. The detailed results for the ablation tests on both the logical inference and the ListOps tasks are found in Table 1 and 3a.

## 4.4 Stanford Sentiment Treebank

The Stanford Sentiment Treebank is a classification task described in Socher et al. (2013). There are two settings: SST-2, which reduces the task down to a positive or negative label for each sentence (the neutral sentiment sentences are ignored), and SST-5, which is a fine-grained classification task which has 5 labels for each sentence.

Current state-of-the-art models use pretrained contextual embeddings Radford et al. (2018); McCann et al. (2017); Peters et al. (2018). Building on ELMO Peters et al. (2018), we achieve a performance

Table 3: Accuracy results of models on the SST.

| | SST-2 | SST-5 |
|---|---|---|
| *Sequential sentence representation & other methods* | | |
| Radford et al. (2017) | 91.8 | 52.9 |
| Peters et al. (2018) | – | 54.7 |
| Brahma (2018) | 91.2 | 56.2 |
| Devlin et al. (2018) | 94.9 | – |
| Liu et al. (2019) | 95.6 | – |
| *Recursive NN + ground-truth structure* | | |
| Tai et al. (2015) | 88.0 | 51.0 |
| Munkhdalai and Yu (2017) | 89.3 | 53.1 |
| Looks et al. (2017) | 89.4 | 52.3 |
| *Recursive NN + latent / learned structure* | | |
| Choi et al. (2018) | 90.7 | 53.7 |
| Havrylov et al. (2019) | 90.2±0.2 | 51.5±0.4 |
| Ours (Glove) | 90.4 | 52.2 |
| Ours (ELMO) | 92.0 | 55.2 |

comparable with the current state-of-the-art for both SST-2 and SST-5 settings. However, it should be noted that our model is a sentence representation model. Table 3 lists our and related work's respective performance on the SST task in both settings.

## 5 Conclusion

In this paper, we introduce the Ordered Memory architecture. The model is conceptually close to previous stack-augmented RNNs, but with two important differences: 1) we replace the pop and push operations with a new writing and erasing mechanism inspired by Ordered Neurons (Shen et al., 2018); 2) we also introduce a new Gated Recursive Cell to compose lower level representations into higher level one. On the logical inference and ListOps tasks, we show that the model learns the proper tree structures required to solve them. As a result, the model can effectively generalize to longer sequence and combination of operators that is unseen in the training set, and the model is data efficient. We also demonstrate that our results on the SST are comparable with state-of-the-art models.

## Footnotes

[3]All parsing scores are given by `Evalb https://nlp.cs.nyu.edu/evalb/`

[4]`http://nlp.seas.harvard.edu/2018/04/03/attention.html`

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
