[Supplementary Material]



# A    Tree induction algorithm

**Data:** $p_1, ..., p_T$
**Result: T**
initialize queue = $[w_2, ..., w_T]$
    stack = $[w_1], h = \text{argmax}(p_1) - 1$;
**for** $i \leftarrow 2$ **to** $T$ **do**
    $y_i = \text{argmax}(p_i)$;
    $d = y_i - h$;
    **if** $d > 0$ **then**
        **for** $j \leftarrow 1$ **to** $d$ **do**
            **if** len(stack) $< 2$ **then**
                **Break**;
            **end**
            $e_1 = \text{stack.pop}()$;
            $e_2 = \text{stack.pop}()$;
            $\text{stack.push}(\text{node}(e_1, e_2))$;
        **end**
    **end**
    $\text{stack.push}(\text{queue.popleft}())$;
    $h = y_i - 1$;
**end**
**while** len(stack) $> 2$ **do**
    $e_1 = \text{stack.pop}()$;
    $e_2 = \text{stack.pop}()$;
    $\text{stack.push}(\text{node}(e_1, e_2))$;
**end**

**Algorithm 2:** Shift-reduce parsing algorithm for generate parsing tree from Ordered Memory model. Here we greedily choose the $\text{argmax}(p_t)$ as the head of stack for each slot.

# B    Hyperparameters

Table 4: The hyperparameters used in the various experiments described. $D$ is the dimension of each slot in the memory. There are 4 different dropout rates for different parts of the model: *In* dropout is applied at the embedding level input to the OM model. *Out* dropout is applied at the layers in the MLP before the final classification task. *Attention* dropout is applied at the layers inside stick-breaking attention mechanism. *Hidden* dropout is applied at various other points in the OM architecture.

| Task | Memory size | #slot | Dropout | | | | Batch size | Embedding | |
|------|-------------|-------|---------|---|---|---|------------|------|-----------|
| | | | *In* | *Out* | *Hidden* | *Attention* | | Size | Pretrained |
| Logic | 200 | 15 | 0.2 | 0.2 | 0.2 | 0.2 | 128 | 200 | None |
| ListOps | 128 | 21 | 0.1 | 0.1 | 0.1 | 0.1 | 128 | 128 | None |
| SST(Glove) | 300 | 15 | 0.3 | 0.4 | 0.2 | 0.2 | 128 | 300 | Glove |
| SST(ELMO) | 300 | 15 | 0.3 | 0.2 | 0.2 | 0.3 | 128 | 1024 | ELMo |

# C    Dynamic Computation Time

Given Eqn. 7, we can see that some $o_t^i$s are multiplied with $\overrightarrow{\pi}_t^i$. It may not necessary to compute the cell function (Eqn. 6) if the cumulative probability $\overrightarrow{\pi}_t^i$ is smaller than a certain threshold. This threshold actively controls the number of computation steps that we need to perform for each time step. In our experiments, we set the threshold to be $10^{-5}$. This idea of dynamically modulating the number of computational step is similar to Adaptive Computation Time (ACT) in Graves (2016), which attempts to learn the number of computation steps required that is dependent on the input.

428 However, the author does not demonstrate savings in computation time. In Tan and Sim (2016), the
429 authors implement a similar mechanism, but demonstrate computational savings only at test time.