[Reviews · NeurIPS 2019]

Reviewer 1



This paper presents a novel model design/algorithm for building compositional representations of sequences when (as in natural language or code) it is presumed that the sequences have salient latent structure that can be described as a binary tree. The method performs essentially at ceiling on two existing artificial datasets that were designed for this task, both of which have not been previously solved under comparable conditions. The method also performs reasonably well on a sentiment analysis task. Pros: The method is novel and solves a couple of prominent instances of an important open problem in deep learning for NLP and similar domains with latent structure: How to we build models that can efficiently learn and to build compositional representations using latent structure? This is interesting and likely to garner a reasonably large audience as a somewhat abstract/artificial result. It is also plausible (though definitely not certain) that this models of this kind could yield significant advances in interpretability and robustness in downstream applications. Cons: If I understand correctly, there is a modest limitation of the model that is never really mentioned in the paper: At least at training time, the model takes O(N^2) steps to do O(N) work. A binary tree of N nodes only requires N-1 merge/reduce operations, but in this model, the cell() operation must be applied sequentially N^2 times to fully build an arbitrary tree. Some of this cost could be saved at test time. This should be discussed more in the paper, but it's not fatal: It's better to have a polynomial time algorithm that works than a linear-time one that doesn't. In addition, while the paper doesn't discuss the resources used for training, I expect that this polynomial-time backprop-friendly method is more efficient than linear time models like Yogatama's that must be trained using slow and unstable RL-style mechanisms. Also, as I hinted at above, just because this model can discover and use structure reliably on clean artificial data, it is not obvious that this model will actually be able to discover structure consistently on natural noisy data. I think this is a significant and publishable advance, and I don't see any overt overclaiming, but this is a clear opportunity for improvement. The SST sentiment results are a bit disappointing, both because this is such a straightforward/easy/longstanding task, and because there is no evaluation or visualization of what structure the model uses on that data. The reproducibility checklist suggests that source code is available now, but I can't find any. Nit: Most of the arXiv papers that are cited have been published. Consider citing the published versions.

Reviewer 2



This paper proposes Ordered Memory (OM), a new memory architecture that simulates a stack based on the principles of Ordered Neurons. In practice, OM operates similar to a continuous shift-reduce parser---where given an input sequence OM decides how many reduce operations are performed before shifting the new input into the stack. OM uses a stick-breaking attention method to model the number of reduce steps and a novel Gated Recursive Cell to implement the reduce step. Experiments on logical inference and ListOps show that OM is able to model required tree structures to solve the tasks and generalize to longer sequences that are not seen in training. Experiments on sentiment analysis show that OM achieves good performance on a real-world dataset as well. While OM relies on many principles introduced in Ordered Neuron, I think that it is not straightforward to apply it to memory and therefore is still a good methodological contribution. The writing of the paper can be improved. I have published in this area in the past but it still took me some time to actually understand how the new method operates. I suggest improving the illustration in Figure 1. I also found several typos in the paper (e.g., in Eq. 5). In terms of technical contributions, my major comment about the paper is comparisons with other stack-augmented recurrent neural networks. The authors show that OM outperforms LSTM, RRNet, ON-LSTM, Transformer, and Universal Transformer on the logical inference task. I would think that other stack-based methods (e.g., Joulin and Mikolov, 2015, Yogatama et al., 2018) would be able to perform well on this task as well. Similarly for the ListOps task. What is the main benefit of OM compared to these similar methods?

Reviewer 3



UPDATE: Thank you authors for your response, and for adding the model ablations. I have changed my score to accept, however my concerns still stand regarding the performance of this model on real language tasks. I think the architecture proposed is certainly interesting. However I have some concerns with the evaluation used in this work (namely the method doesn't do well on the sole real language task attempted), and how it does not actually validate the hypothesis (e.g. that this method learns compositionality of real language and as a result outperforms other methods). Moreover, the authors identify two core contributions that are distinct from prior work (stick-breaking attention and gated recursive cell), but do not analyse the effectiveness of these two contributions through any ablation studies. Some other comments on the writing: - The introduction contains a substantial discussion of related works. Personally I find this distracting. Out of the ~45 lines used for the introduction, only ~10 lines actually describe what this work is about. - I think perhaps there is a better title than Ordered Memory, as "ordered memory" doesn't really motivate anything related to compositionality, when compositionality seems to be the main motivation of this paper. - In equation 5, the superscript i for the (1-\pi) term should be inside the bracket)

[Author Response · NeurIPS 2019]

| | Logical Inference | | | | | | ListOps |
|---|---|---|---|---|---|---|---|
| Ordered Memory | $98 \pm 0.4$ | $97 \pm 0.5$ | $96 \pm 0.8$ | $94 \pm 0.8$ | $94 \pm 1.5$ | $92 \pm 0.7$ | $99.9 \pm 0.02$ |
| ~~cell(·)~~ TreeRNN Op. | 69 | 67 | 65 | 61 | 57 | 53 | 63.1 |
| ~~Stick-breaking~~ softmax | 98 | 97 | 96 | 92 | 93 | 92 | 98 |
| Yogatama et al. (2018) | 58 | 57 | 58 | 56 | 55 | 50 | 60 |

Table 1: We replaced the $\text{cell}(\cdot)$ operator with the RNN operator found in TreeRNN, which is the best performing model that explicitly uses the structure of the logical clause. In this test, we find that the TreeRNN operator results in a large drop across the different tasks. We also replaced the stick-breaking process with masked and scaled softmax: $p_t = \text{softmax}\left(\frac{\hat{\beta}_t}{\sqrt{d}}\right)$ where $\hat{\beta}_t$ is defined in Section 3.1 and $d$ is the dimension of memory slot. The purpose of this is to scale down the logits before softmax is applied, a technique similar to the one seen in Vaswani et al. (2017). Surprisingly, we observed that the masked and scaled softmax results in a more robust model (the model is less sensitive to hyper-parameters, and thus easier to train) while the stick-breaking formulation provides marginally better performance. The reason could be that softmax is more numerically stable for both feedforward and backpropagation. As discussed in Section 3.3 of the paper, the stick-breaking formulation was initially used to reflect the process that a shift-reduce parser would make if the decisions were made one after another.

Thank you all for your detailed review and insightful comments.

Firstly, to address comments on using natural language data: We have indeed found it challenging to learn structure from real language data and associated tasks. We think that a natural language task with a more informative signal (perhaps language modelling) would be able to correct this. This will be the direction in which we take our future work.

We have conducted an ablation test for the Gated Recursive Cell and Stick-breaking Attention. Results of ablation test are shown in above table. We will add more discussion about different attention methods to our paper.

**Reviewer 1**  You are correct regarding complexity of the model during training time. We will include a description of this in the camera-ready version. We will also fix the bibliography to reflect the conferences/proceedings the arxiv-ed papers were published in.

As for the reproducibility checklist, we thought that checking that box meant that we would release the code *after* the paper has been published.

**Reviewer 2**  We have actually included the performance of models that learn a tree structure. For ListOps, we also have also listed the results for RL-SPINN, which learns to use a stack. In addition, we have also tested our implementation of the stack-augmented model in Yogatama et al. (2018). We currently have preliminary results using that model (See the above table for detailed results on Logical Inference and ListOps). Note that these results are expected to change as we find better hyperparameters for this model.

We will correct the typos in the paper for the camera-ready version. Our apologies for not catching them before submission.

**Reviewer 3**  We find that it is difficult to show that our model learns compositionality on real language due to the lack of datasets that explicitly test for this property. The various toy tasks that we have tested our model on were designed to isolate this capability, and so we have used them to demonstrate this to the extent that we can. And because we know the structure of the data in the logical inference task, we were also able to remove clauses from training in order to see how well the model generalises when tested on them during evaluation. In those cases, to generalise to those unseen substructures requires compositionality.

We understand your criticism with respect to ablation studies. We have provided the details of the ablation studies above, and the detailed results are shown in the table.

Also, we apologise for the lengthy discussion of related work, as we thought providing a comprehensive coverage of existing work would show the state of the field much more clearly. We will make amendments to the related work section as we accommodate all the reviewer comments in our camera-ready version.

**References**

Ashish Vaswani, Noam Shazeer, Niki Parmar, Jakob Uszkoreit, Llion Jones, Aidan N Gomez, Łukasz Kaiser, and Illia Polosukhin. 2017. Attention is all you need. In *Advances in Neural Information Processing Systems*, pages 5998–6008.

Dani Yogatama, Yishu Miao, Gabor Melis, Wang Ling, Adhiguna Kuncoro, Chris Dyer, and Phil Blunsom. 2018. Memory architectures in recurrent neural network language models.


[Meta-Review · NeurIPS 2019]

The reviewers reached, after discussion, the consensus that this paper presenting a novel way of modelling strucutured memory is worth including in the conference. The modelling aspect of the paper was of interest to the reviewers, who were furthermore reasonably confident that the method has empirical merit thanks to the experiments both synthetic and "real world". Perhaps the main weakness of this paper is that while the synthetic experiments prove the concepts and the sentiment analysis experiments show robustness to noisy data, further non-synthetic experiments might have further showcased applications of this method to tasks which the community cares about. For now, I find it of a sufficient standard for publication, and anticipate that further work will demonstrate whether this method stands well against other tasks... or not.